# Sample-efficient policy learning in multi-agent Reinforcement Learning via meta-learning

## Abstract

To gain high rewards in muti-agent scenes, it is sometimes necessary to understand other agents and make corresponding optimal decisions. We can solve these tasks by first building models for other agents and then finding the optimal policy with these models. To get an accurate model, many observations are needed and this can be sample-inefficient. What's more, the learned model and policy can overfit to current agents and cannot generalize if the other agents are replaced by new agents. In many practical situations, each agent we face can be considered as a sample from a population with a fixed but unknown distribution. Thus we can treat the task against some specific agents as a task sampled from a task distribution. We apply meta-learning method to build models and learn policies. Therefore when new agents come, we can adapt to them efficiently. Experiments on grid games show that our method can quickly get high rewards.

## 1 Introduction

Applying Reinforcement Learning (RL) to multi-agent scenes requires carefully consideration about the influence of other agents. We cannot simply treat other agents as part of the environment and apply independent RL methods (Lanctot et al., 2017) if the actions of them has impact on the payoff of the agent to be trained. For example, consider the two-person ultimatum bargaining game, where two players take part in. One player propose a deal to split a fixed amount of money for them two and the other player decides to accept it or not. If the second player accepts the proposal, they split the money, but if the proposal is refused, they both get zero. Experimental results (Güth et al., 1982) show that in actual life, the second player makes the decision according to whether he or she judge the final result fair, rather than makes the obvious rational decision. Thus, the first player needs to predict how the second player will react so as to make the proposal acceptable.

In order to exploit the other agents and find the corresponding optimal policy, we need to understand these agents. Here in this paper, we call all the other agents "opponents" to distinguish our agent from them, even if they may have cooperative relationship with our agent. For simplicity, we only consider tasks with only one opponent. Extension to tasks with more opponents is straightforward. A general way to exploit an opponent is to build a model for it from observations. This model can characterize any needed feature of the opponent, such as next action or the final goal. Such a model can make predictions for the opponent and thus turns the two-agent task into a simple-agent decision making problem. Then we can apply various RL methods to solve this problem.

It is necessary that we need to have an accurate model for the opponent to help make decision. Previous works (He et al., 2016; Raileanu et al., 2018) propose some methods to model the opponent. Generally, it requires many observations to get a precise model for the opponent. This may cost many iterations to act with the opponent. What's more, even if we can precisely model the opponent, there exists a main drawback of above process that the performance of the learned policy has no guarantee for any other opponent. Things are even worse if opponents have their private types which are unknown for us. New opponents with different types can have different policies or even different payoffs. Therefore, it seems that when a new opponent came, we have to learn a policy from the beginning. In some practical situations, the whole opponents follow a distributions over all these possible types. Let's come back to the ultimatum bargaining game. Bahry & Wilson (2006) shows

that people with different ethnicity may have different standards for fairness. Thus if we assume the type for player 2 to be its judgment for fairness, there can be a distribution for types dependent on the ethnic distribution. Given that opponents follows a distribution, it is possible that we can employ some given opponents to help us speed up the process of opponent modeling and policy improving for the current opponent.

If we consider the policy learning against a specific opponent as a task, our goal can be considered as training a policy on various tasks so that it can efficiently adapt to a good policy on a new task with few training samples. This is exactly a meta-learning problem. We employ Model-Agnostic Meta-Learning (MAML)(Finn et al., 2017) to conduct meta-learning. Rabinowitz et al. (2018) applied meta-learning to understand opponents, but this work doesn't address the policy improvement for the agent to be trained. We apply meta-learning to opponent modeling and policy learning separately while training the two meta-learners jointly. Then we use the meta-learners to initialize the model and policy for the new opponent. Experimental results show that the agent can adapt to the new opponent with a small number of interactions with the opponent.

## 2 PRELIMINARY

In this section, we introduce some preliminaries of our work. We formalize our task based on stochastic games(Littman, 1994), which is a general framework for multi-agent problems, and Bayesian games(Harsanyi, 1967), which formalize the incomplete information of players. Next we introduce Model-Agnostic Meta-Learning (MAML), a meta-learning algorithm that is applicable to various gradient-based methods. Our approach employs MAML as the meta-learning method to train across tasks.

### 2.1 FORMALIZATION

We first introduce stochastic games. Formally, a stochastic game, with $N$ players, is defined as $\langle N, S, \{A_1, ..., A_N\}, T, \{R_1, ..., R_N\}\rangle$, where $S$ is the set of states, $A_i$ is the action set for player $i$, $R_i : S \times A_1 \times ... \times A_N \to \Delta([0,1])$ is the reward function for player $i$ ($\Delta(C)$ denotes the probability distributions over a set $C$), and $T : S \times A_1 \times ... \times A_N \to \Delta(S)$ is the transition function. The goal for player $i$ is to maximize its own cumulative reward $\mathbb{E}[\sum_{t=0}^{T} \gamma^t r_{i,t}]$, where $r_{i,t}$ is the reward player $i$ gets at time step $t$ and $\gamma$ is the discounting factor. $\pi_i : S \to \Delta(A_i)$ denotes the policy for player $i$, which maps each state to a distribution over actions.

Our agent takes the role of one player. With out loss of generality, we assume that our agent is player 1. In this paper we only consider two-player stochastic games.

Bayesian games further introduce types for players. Since our agent takes the role of player 1, the different types of player 1 can be considered as different states. We only consider that player 2 has a set of types $\Theta$. A specific opponent playing player 2 has its own $\theta \in \Theta$ which is unknown to player 1. There exists a prior distribution $p(\theta)$ over the population of opponents. Under this setting, each $\theta \in \Theta$ has its corresponding reward function $R_i(s, a_1, a_2, \theta)$ for each $i \in \{1, 2\}$, $s \in S$, $a_1 \in A_1$ and $a_2 \in A_2$.

Therefore combining the above concepts we formalize our tasks as Bayesian stochastic games $\langle N, S, \{A_1, ..., A_N\}, T, \{R_1, ..., R_N\}, \Theta, p\rangle$, where $R_1, ..., R_N$ are the modified reward functions dependent on $\theta \in \Theta$ and $p$ is the prior distribution over $\Theta$ for types of player 2.

### 2.2 MODEL-AGNOSTIC META-LEARNING

Meta-learning considers the goal that aims to quickly train a model for a new task with the help of data on many similar tasks. Formally, we denote $\{\mathcal{T}\}_{i=1}^{N_1}$ the given $N_1$ tasks used for training. Then $N_2$ more new tasks $\{\mathcal{T}\}_{i=N_1+1}^{N_1+N_2}$ are used for testing. In the meta-learning setting, we assume that the tasks are sampled from a distribution $p(\mathcal{T})$ over all possible tasks. For each task $\mathcal{T}_i$, we want to identify a mapping $f_i$, that maps each input $\mathbf{x}$ to its corresponding output $\mathbf{y}$.

Model-Agnostic Meta-Learning (MAML) is one of the best meta-learning algorithms that can be applied to models that are trained with gradient descent. Denote the parameter for each $f_i$ as $\psi_i'$.

The loss for $f_{\psi_i'}$ on task $\mathcal{T}_i$ is denoted as $\mathcal{L}_{\mathcal{T}_i}(f_{\psi_i'})$. A meta-learner with parameter $\psi$ is used as the initialization for all $\psi_i'$s and the update for each specific task is:

$$\psi_i' = \psi - \alpha \nabla_\psi \mathcal{L}_{\mathcal{T}_i}(f_\psi),$$

where $\alpha$ is the learning rate. The update for $\psi$ is:

$$\psi = \psi - \beta \nabla_\psi \sum_{i=1}^{N_1} \mathcal{L}_{\mathcal{T}_i}(f_{\psi_i'}),$$

where $\beta$ is the learning rate for $\psi$.

## 3 OUR APPROACH

Before we dive into the Bayesian stochastic games, we first consider the stochastic game that we have a specific opponent with type $\theta \in \Theta$. We aim to explicitly model the opponent so that we can get predictions about it. The predicted value should be some character of the opponent that can help our agent improve policy. For example, in games where our reward is related to the final goal of the opponent, we can directly predict its goal. Formally, at some state $s_t \in S$, our agent predicting some estimated value $\tilde{v}(\cdot|s_t, \theta)$, where $\tilde{v}$ is the estimation of value $v$. $v$ represent a character of the opponent such as goals, next actions or next positions. For convenience, we use $v_\theta$ to denote $v(\cdot|\theta)$. Then our agent can choose an action according to $\pi_1(\cdot|s_t, \tilde{v}_\theta(s_t))$, while an agent unaware of its opponent can only take action according to $\pi_1(\cdot|s_t)$. Thus, the task has been divided into two sub-tasks: modeling $\tilde{v}_\theta$ of the opponent and learning a policy $\pi_1(\cdot|\tilde{v}_\theta)$. The latter task can be considered as a general RL problem and we can apply various RL methods to solve it.

Now assume that opponents have a prior distribution for their types. That is, an opponent with type $\theta \in \Theta$ can be treated as a sample following $\theta \sim p(\theta)$. If we can collect data from some opponents sampled from $p$, it is possible we can generalize the model and policy to the game with a new opponent. Thus we can consider the opponent modeling and policy learning as two meta-learning tasks. The former can be considered as an imitation learning or supervised learning task while the second is a RL task. Both can apply MAML for meta learning. Since the learned policy needs the model to make prediction, we cannot train the two meta-tasks independently. We jointly training the model and policy with some given opponents. We call our method as Meta-Opponent-Agent learning (MOA) to indicate that the model and policy are jointly trained via meta-learning. Finally when a new opponent comes, we initialize our model and policy with the meta-model and meta-policy. The training procedure is shown in the figure 1.

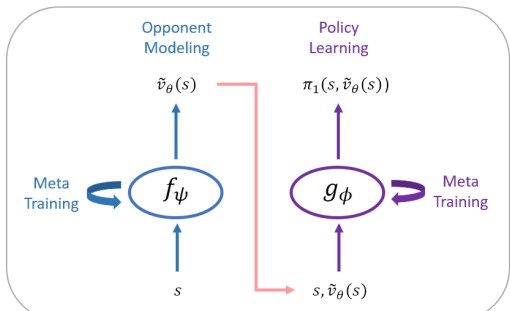

Figure 1: Training Procedure: Network $f_\psi$ for opponent modeling takes state $s$ as input and outputs the prediction value $\tilde{v}_\theta(s)$, where $\theta$ is the private type for current opponent. Network $g_\phi$ for policy learning takes state $s$ and $\tilde{v}_\theta(s)$ as input and out put the policy. Both networks are train via MAML.

### 3.1 OPPONENT MODELING

The opponent in our game is considered as some player that won't adapt its policy to our agent. This assumption can be true in many practical situations. For example, the consumers of a business

man usually have stable preferences. Then we further assume that there exist a distribution over the policies of the opponents. This assumption can also be considered true for the business man situation, where a specific consumer is just a sample from the whole population. Our goal here is to model the current opponent with the help of data that are collected by playing with other opponents.

Formally, we now aim to model the value function $v$ for the opponent from the observations. As mentioned above, $v$ here can represent many forms of characters of the opponent, such as the final goal, next action, next position or any other character we wish to predict. For each type $\theta \sim p$, the value function is specified as $v_\theta$. Now we have $M$ opponents sampled from the population. Opponent $i$ has a type $\theta_i \sim p(\theta)$. For any given state $s$ as an input for the opponent $i$, the model outputs $\tilde{v}_{\theta_i}(s)$. The task of minimizing $\tilde{v}_{\theta_i}$ and $v_{\theta_i}$ are treated as task $\mathcal{T}_i^o = \{(s, v_{\theta_i}(s)\}$ with loss function $\mathcal{L}_{\mathcal{T}_i^o} = dist(v_{\theta_i}, \tilde{v}_{\theta_i})$, where $dist(\cdot, \cdot)$ is the distance metric for two $v$ functions. The distance function can vary for different problems. Following the framework of MAML, we use a network called opponent network (OPNet) $f$ with parameter $\psi$ to learn the function $\tilde{v}$. Then for each opponent, we collect data $S_i = \{(s, v_{\theta_i}(s)\}$ and then use this dataset to update $\psi$ to get the adapted parameter $\psi_i'$. New dataset $S_i' = \{(s, v_{\theta_i}(s)\}$ is collected with $f_{\psi_i'}$. $\{S_i'\}_{i=1}^M$ are together used to update $\psi$. Finally we use the learned $\psi$ to initialize the parameter for the current opponent as $f_\psi$. The updating process for parameters follows the framework of MAML.

Following Grant et al. (2018), we are learning an empirical Bayesian model over the opponent population after training with the $M$ opponents. When data for the new opponent are observed, it is easier to adapt to the new opponent with such a prior model.

## 3.2 POLICY LEARNING FOR THE AGENT

With a model of the opponent, our agent can give a better policy than the policy unaware of the opponent. The policy learning for our agent is also trained by a meta-learning process, similar to the above opponent modeling process. To distinguish the notation for opponent modeling, we use $\phi$ to represent the parameter of agent's policy. The agent can employ various RL methods to learn the policy. We use the Dueling DQN (Wang et al., 2016) as our learning method. We use $g_\phi$ to denote the Dueling DQN mapping with parameter $\phi$. The $M$ opponents bring $M$ meta-training tasks $\{\mathcal{T}_i^a\}_{i=1}^M$. For opponent $i$, at state $s_t$, OPNet predict a value $f_\psi(s_t)$. The policy $\pi_i^a$ of our agent is defined on the new state $(s_t, f_\psi(s_t))$ and the action $a_t$ the agent chooses is sampled from $\pi_i^a(s_t, f_\psi(s_t))$. Then the agent gets a immediate reward $r_t$. We collect a dataset $D_i = \{(s_t, f_\psi(s_t), a_t, r_t)\}$ for the task $\mathcal{T}_i^a$. Similar to the above part, we update $\phi$ with $D_i$ to get adapted parameter $\phi_i'$. Next, we use $f_{\psi_i'}$ and $\gamma_{\phi_i'}$ to get dataset $D_i' = \{(s_t, f_{\psi_i'}(s_t), a_t, r_t)\}$. Parameter $\phi$ is in return updated with $\{D_i'\}_{i=1}^M$. When finally the agent meet the new opponent, it uses $g_\theta$ as the initialization for its policy and improve its policy.

From the Bayesian point of view, we can consider the learned $g_\phi$ is the approximated Bayesian Optimal Policy against the opponent distribution. When we meet a new opponent, we initialize the policy with $g_\phi$ to accelerate the learning by guiding the agent to explore the potential direction.

## 3.3 ALGORITHM

We train $f_\psi$ and $g_\phi$ jointly, since the prediction of the opponent is considered as part of the input for the agent. More concretely, for each iteration, our agent play with each opponent $i \in [M]$. Our agent use the OPNet to predict the opponent, and the Dueling DQN use the prediction as part of its input to give a policy. Then both OPNet and Dueling DQN are updated. The algorithm is shown in Algorithm 1.

## 4 RELATED WORK

Many works concentrate on multi-agent Reinforcement learning tasks. Works like Lanctot et al. (2017) connect these tasks with game theory to modify general RL methods to multi-agent scenes. Some works aims to solve equilibriums for specific games. For example, Silver et al. (2017) proposes a self-play deep RL method for two-player zero-sum perfect-information games and their work on Go outperforms human beings. Some other researches aims to learn policies for agents via

---

**Algorithm 1** Meta-Opponent-Agent Learning

---

**Require:** $p(\theta)$ over types of opponents
**Initialization:** Get $M$ opponent samples $\mathcal{T}_i^o$, each with type $\theta_i \sim p(\theta)$. Initialize $f_\psi$, $g_\phi$.
**repeat**
    **for** all $\mathcal{T}_i^o$ **do**
        Agent use $f_\psi$ and $g_\phi$ to play with opponent $i$. Collect data $S_i = \{(s_t, v_{\theta_i}(s_t))\}$ and $D_i = \{(s_t, f_\psi(s_t), a_t, r_t)\}$.
        Evaluate $\nabla_\psi \mathcal{L}_{\mathcal{T}_i^o}(f_\psi)$ and $\nabla_\phi \mathcal{L}_{\mathcal{T}_i^a}(g_\phi)$ using $S_i$.
        Compute parameters $\psi_i' = \psi - \alpha \nabla_\psi \mathcal{L}_{\mathcal{T}_i^o}(f_\psi)$ with $S_i$ and $\phi_i' = \phi - \alpha \nabla_\phi \mathcal{L}_{\mathcal{T}_i^a}(g_\phi)$ with $D_i$.

        Agent use $f_{\psi_i'}$ and $g_{\phi_i'}$ to play with opponent $i$. Collect data $S_i' = \{(s_{,t}, v_{\theta_i}(s_t))\}$ and $D_i = \{(s_t, f_{\psi_i'}(s_t), a_t, r_t)\}$.
    **end for**
    Update $\psi = \psi - \beta \nabla_\psi \sum_{\mathcal{T}_i^o} \mathcal{L}_{\mathcal{T}_i^o}(f_{\psi_i'})$ with $S_i'$ and $\phi = \phi - \beta \nabla_\phi \sum_{\mathcal{T}_i^a} \mathcal{L}_{\mathcal{T}_i^a}(f_{\phi_i'})$ with $D_i'$
**until** Done

---

imitation learning (Oh et al., 2014; Thurau et al., 2004). These works just wish to identify good polices and don't aim to exploit specific opponents.

There are also some works address opponent modeling. Raileanu et al. (2018) proposes a method to automatically infer the goal of others by using agent itself. However, this work is not suitable for games that are not goal-directed. What's more, concentrating on specific opponents can lead to a weak policy against other opponents. Works like Johanson et al. (2008); Ganzfried & Sandholm (2015) attempt to model specific opponents while learning a robust policy. These works address the problem from the game-theoretical point of view. They don't have assumptions on opponents.

If we consider the opponent population and assume that there are some prior distribution over the policies of opponents, we are in fact easy to infer information for the current opponent with the help of other opponents. Rabinowitz et al. (2018) connects opponent modeling with theory of mind, a concept from psychology. This work use meta-learning to build flexible and sample efficient models for opponents. However, it ignores the the process for policy learning. Our work attempts to gain information for both opponent modeling and policy improving from the given opponents.

## 5 EXPERIMENTS

In this section, we test our method on three different kinds of two-player games, each with some specific uncertainty:

- Chasing game: a game where the opponent has a private type set with finite elements;
- Blocking game: a game where the opponent has a private type set with infinite elements;
- Recommending game: a game where the opponent has private type set with infinite elements and the agent has random reward functions.

All these games are grid games where both player 1 and 2 choose a one-step direction as its action. In the grid world, each action can only move to the grid next to it or stay at its current position for one step. Since all these games are based on grid worlds, we choose the value function for the opponent as the goals or the next position of the opponent. Thus, we choose cross entropy as $dist(v_\theta, \tilde{v}_\theta)$ in section 3.1. For each game, we test our method MOA and three other baseline methods. We first introduce these baselines and show the experimental results for each game. Further, in the design of games, we don't set the rewards bounded by $[0, 1]$. We make rewards larger to make it easy to train.

### 5.1 BASELINE METHODS

Meta-Opponent (MO): For this method, we only train the model for opponents when playing with the training opponents. Then we use this model to initialize our model for the new opponent and our agent directly learn from the beginning. More concretely, we only train the parameter $\psi$ and

then train $\psi'_{M+1}$ and $\phi'_{M+1}$ for the new opponent $M+1$. This method is used to show that whether training the agent via MAML can help the agent more efficiently adapt to a new opponent.

Meta-agent without model (MA): We don't model the opponents in this method. Dueling DQN is used to directly learn $\pi^a(\cdot|s)$ for $s \in S$ via MAML. This method is used to show the effect of models.

No Meta-Learning (NM): To demonstrate that meta-learning can do take advantage of the information from other opponents and learn a good policy with fewer samples, we directly train the model and the agent for the new coming opponent.

## 5.2 CHASING GAME

In the chasing game, a grid board of size $8 \times 8$ is given. as shown in figure 2a. Player 1 is represented as the red grid and player 2 is the green one. Player 2 has a private goal, which can be considered as its specific type and is unknown to player 1. The goal is one specific gird on the map. Each player 2 has a specific goal, while the player 2 population have a distribution over the 64 grids. That is, the set of types is a finite set. In this chasing game, the grids that are close to the top left corner are preferred. The probability of the goal's location over the map is visualize in figure 2b. The rule for the game is as follows. Both players takes actions simultaneously. One game lasts for at least 15 steps. When a game begin, player 2 will go directly to its goal and stops there. The only way for player 1 to get rewards is to chase player 2 to its goal before the game ends. If player 1 finds player 2, it gets a reward of 10 and the game ends. If it is at one of the 8 neighborhood grids of player 2 at the end of the game, it gets a reward 5. Otherwise, it has reward 0.

We test the four methods, MOA, MA, MO and NM. For MOA, MA and MO, they all require the meta-training process. 20 opponents are sampled as the meta tasks. Each method trains 800 iterations to get the meta learners and use them to initialize their networks. Then 10 new opponents are sampled as testing tasks. Four methods all train 4000 games for each testing task. We compare their performance along the testing process by averaging the rewards on the 10 testing tasks. In this game, the type for player 2 is its goal and we directly model the goal for player 2.

Figure 2c gives the results of four methods during the testing process. We plot the average rewards over 10 opponents. It is easy to see that MOA outperforms the other methods. Notice that the reward trend for MOA first drops and then raises as the testing process goes on. This shows the process that the meta-learner adapt to the current task. Intuitively, the meta-model would first update itself to the current opponent. Then the meta-policy would improve itself to fit the model. NM learns the testing task without meta-training for neither the model nor the policy. It can still improve its policy but cost many more games than MOA. The comparison of MOA and NM shows that we can gain benefits by training across opponents via meta-learning. MO method just train the meta-model. The result shows that MO just performs similar to NM. This result show that simply training the model of opponent cannot help improve efficiency. MA performs the worst among all these methods, even though it has a meta-training process. This is because it doesn't build a model for the opponent. Ignoring the existence of the opponent just results in the failure to improve policy.

## 5.3 BLOCKING GAME

In blocking games, as shown in figure 3a, has a 9*7 size map. In the initial state, player 1 is the red grid and player 1 is the green one. The goal for player 2 is to pass one of the five ways to reach the top two rows, while the goal of the player 1 is to block player 2 to reach the goal area. There are 5 paths that player 2 can pass to get the goal area and the type for player 2 is the probability of choosing each path. Thus the type set for player 2 is a simplex, which has infinite elements. Each path has only one exit. If player 1 can block player 2 at the exit, player 1 gets a reward 10. Otherwise player 2 will pass the exit and reward 1 will get -10.

The training setting for blocking games is a bit different from chasing games. Only 15 opponents are sampled as the meta tasks. Each opponent has a distribution over 5 path and it samples one path for one game. The prior distribution for opponent's type is the Dirichlet distribution with five $0.5$ as parameters. Each method trains 800 iterations to get the meta-parameters. Then 10 new opponents are sampled as testing tasks. Four methods all train 4000 games. In this game, it is hard to directly model the types of opponents as the value thus we simply choose its next position as the value.

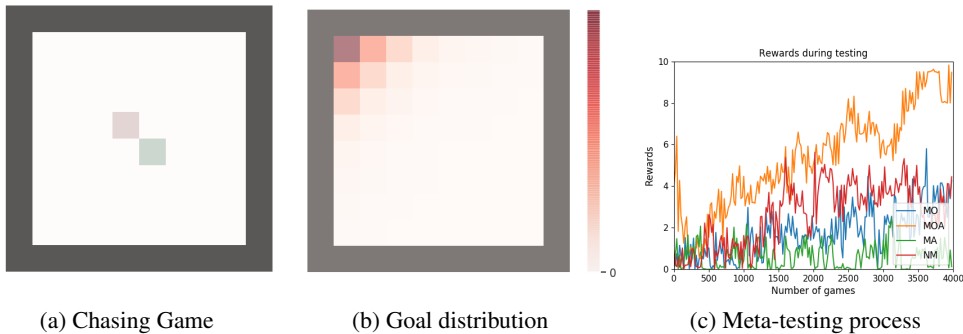

(a) Chasing Game     (b) Goal distribution     (c) Meta-testing process

Figure 2: The results for chasing games. (a) is the initialization state for chasing games. Player 1 is the red grid and player 2 is the green one. Each player 2 has a private goal which is unseen on the map. (b) shows the probability that player chooses its goal. (c) shows the average rewards player 1 gets during meta-testing process. MO, MA and MOA initialize their parameters with results from meta-training process. New opponents are used for testing.

Figure 3b shows the performance of MA and MOA along the meta training process. After 50 iterations, we collect the rewards of our agent gets with the 15 training opponents. Since our agent plays a random policy, we average the rewards against each opponent of 100 games. Notice that MO only has its model trained in the training process, we don't test it. The result shows that MOA can improve itself quickly while MA can hardly improve. This again demonstrate the importance of opponent modeling.

Figure 3c gives the rewards along testing process. It is easy to see that MOA can use less than 500 games to adapt to the new opponents while MO and NM improves slowly. Again MO and NM performs similarly. The results are similar to that of chasing games.

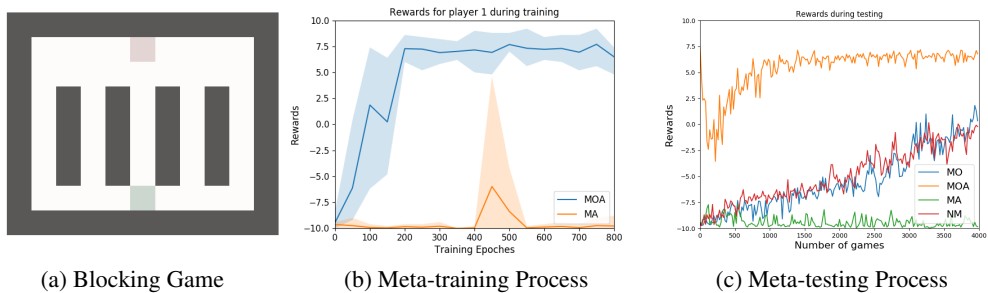

(a) Blocking Game     (b) Meta-training Process     (c) Meta-testing Process

Figure 3: The results for blocking games. (a) is the initialization state. Player 1 is the red one and player 2 is the green one. Each player 2 choose one path to get to the top space and player 1 tries to block it. (b) shows the average rewards player 1 gets during the meta-training process. (c) shows the average rewards player 1 gets during meta-testing process. MO, MA and MOA initialize their parameters with results from meta-training process. New opponents are used for testing.

## 5.4 RECOMMENDING GAME

A recommending game, as shown in figure 4a, has a 7*7 size map. Player 1 is red and player 2 is green. There are 4 blue grids on the left of the map, which are goals for player 2. There are also 4 purple girds on the right of the map, which are objects for player 1. This game is similar to the process that a business man recommends goods for his current consumer. In this game, player 2 also has a private distribution over the 4 goals. The distribution is considered as the type and the prior distribution is a Dirichlet distribution with four $0.5$ as parameters. Player 2 samples a goal from its type distribution and goes to its goal directly. Player 1 needs to recommend one of the

Table 1: Rewards for policies after meta-training (MOA, MO and MA) against 100 new opponents

| Methods | Chasing game | Blocking game | Recommending game |
|---------|--------------|---------------|-------------------|
| MOA | 8.70 | 6.00 | 9.25 |
| MO | 0.00 | -9.6 | 0.00 |
| MA | 4.70 | -10.00 | -0.08 |
| NM | 0.90 | -9.00 | 2.72 |

4 purple objects to player 2. When player 1 reaches one of the objects or the game is played 16 steps, the game ends. Player 1 only gets rewards when it reaches an object. Assume that the vertical coordinate for the goal of player 2 is $y_2$ and that of the recommended object is $y_1$. Then the reward for player 1 is a sample from Gaussian distribution $\mathcal{N}(\mu, 1)$, where $\mu = 10 - 3/2 * |y_1 - y_2|$.

Details of experiments is almost the same as blocking games, except that we choose player 2's goal as its predicting value. Figure 4b demonstrate again that MOA performs well during meta-training. Figure 4c shows that MOA outperforms the other three methods during the testing process. MOA is indeed sample-efficient. The results for recommending games is similar to chasing games and blocking games. The random rewards just bring more variance to the training and testing process.

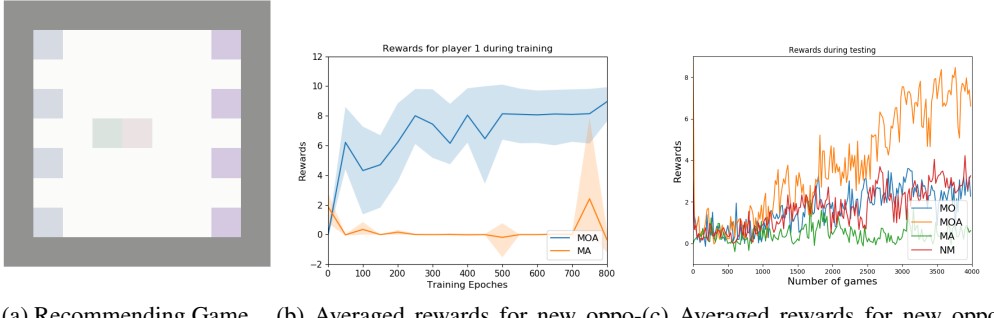

(a) Recommending Game    (b) Averaged rewards for new oppo- (c) Averaged rewards for new opponents during training process    nents during testing process

Figure 4: The results for recommending games. (a) is the initialization state. Player 1 is the red grid and player 2 is the green one. Blue grids are goals for player 2 and purple grids are objects for player 1. Each player 2 choose one of goals and player 1 tries to recommend a corresponding object. (b) shows the average rewards player 1 gets during the meta-training process. (c) shows the average rewards player 1 gets during meta-testing process. MO, MA and MOA initialize their parameters with results from meta-training process. New opponents are used for testing.

Finally we test all the four methods on 100 opponents sampled from the opponent distribution. For each opponent, we just play one game with it. That is, we don't conduct any learning process for new opponents. The results for all three games are given in the table 1. As the table shows, MOA can reaches relatively high rewards while other methods performs bad. The results demonstrate that MOA can gain prior information from meta-learning process.

## 6 CONCLUSION

In the face of other agents, it is beneficial to build models for opponents and find a corresponding good policy. This method can be sample-inefficient since it costs many observations to build models and learn a policy then. We propose a method that can employ the information learned from experiences with other opponents to speed up the learning process for the current opponents. This method is suitable for many practical situations where the opponent population has a relative stable distribution over their policies. We apply meta-learning to jointly train the opponent modeling and policy improving process. Experimental results show that our method can be sample-efficient.

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
