# OpenReview forum: "Sample-efficient policy learning in multi-agent Reinforcement Learning via meta-learning"
_ICLR.cc/2019/Conference_

### Official Review · AnonReviewer2 · 2018-10-31
**Overall interesting idea, unclear on technical details, missing an important baseline.**

**Rating:** 4
**Confidence:** 4

**Review:**

This paper proposes to apply MAML to a multi-agent setting. In this formulation each opponent corresponds to a task and two separate parts of the policy are learned via meta-learning:
1) the opponent modelling network that predicts the value function for a given opponent based on past actions and states.
2) the policy network which takes in the state and the predicted value function of the opponent.
The main concern with this paper is the lack of technical detail and an important missing baseline. The paper also suffers from lacking clarity due to a large number of grammatical mistakes.

Technical detail and concerns:
The paper mentions Duelling DQN as the RL algorithm in the inner loop. This is very unusual and it's a priori unclear whether MAML with DQN in the inner loop is a sensible algorithm. For example, DQN relies both on a target network and an argmax operator which seem to violate the differentiability requirements needed for MAML regarding higher order gradients. The authors entirely miss this and fail to address possible concerns.

The authors also fail to provide any details regarding the exploration scheme used. In fact, a value function is never mentioned, instead the authors talk about a policy pi^a_i, leaving it unclear how this policy is derived from the value function. When the Q-function takes as input the true opponent, there is no need for meta-learning of the policy: Given a known opponent, the tuple (s_t, opponent) defines a Markov state. As far as I could gather from the paper, the authors are missing a baseline which simply learns a single Q-function across all opponents (rather than meta-learning it per opponent) that takes as input the predicted opponent.
My expectation is that this is more or less what is happening in the paper. The authors also fail to compare and contrast their method to a number of recent multi-agent algorithms, eg. MADDPG, COMA and LOLA.

Furthermore, the results are extremely toy and seem to be for single runs , rendering them insignificant.

While the idea itself is interesting, the above concerns render the paper unsuitable for publication in it's current form.

---

### Official Review · AnonReviewer3 · 2018-11-12
**interesting idea, but the experiments do not validate the approach for opponent modeling**

**Rating:** 4
**Confidence:** 4

**Review:**

The paper presents an approach to multi-agent learning based on the framework of model-agnostic meta learning. The originality of the approach lies in the decomposition of the policy in two terms, with applications to opponent modeling: the first part of the policy tries to predict some important characteristic of the agent (the characteristic itself is prior knowledge, the value it takes for a particular opponent is learnt from observations). The second part of the policy takes the estimated characteristic of the opponent as input, the current state and produces the action. All networks are trained within the MAML framework. The overall approach is motivated by the task of opponent modeling for multi-agent RL.

The approach makes sense overall -- the "value" of the opponent is valuable prior knowledge. The originality is limited though. In this kind of paper, I would expect the experiments to make a strong case for the approach. Unfortunately, the experiments are extremely toyish and admittedly not really "multi-agent": the "opponent" has a fixed strategy that does not depend on what the current is doing (it is therefore not really an opponent). The experimental protocol is more akin to multitask RL than multi-agent RL, and it is unclear whether the approach could/should work for opponent modeling even on tasks of low complexity. In other words, the experimental section does not address the problem that is supposed to be addressed (opponent modeling).

other comments:
- "The opponent in our game is considered as some player that won’t adapt its policy to our agent." -> in the experiments it is worse than that: the opponents actions do not even depend on what the agent is doing... So admittedly the experiments are not really "multi-agent" (or "multi-agent" where the "opponent" is totally independent of what the agent is currently doing).

- "Each method trains 800 iterations to get the meta learners and use them to initialize their networks. Then 10 new opponents are sampled as testing tasks. Four methods all train 4000 games for each testing task." -> what does 800 iterations mean? Does it mean 800 episodes (it would seem strange for a "fast adaptation task" to have fewer episodes for training than for testing).

- "Notice that the reward trend for MOA first drops and then raises as the testing process goes on. This shows the process that the meta-learner adapt to the current task." -> the adaptation to the new opponent does not really explain the drop?

- Figure 3(c): the MA baseline has a reward of ~-10, which is worse than random (a uniform random placement at the 5 strategic positions would get 10*1/5-10*4/5 = -6). On the other hand, MOA achieves very high rewards, which indicates that the "opponents" strategies have low entropy. What is the best achievable reward on the blocking game?

---

### Official Review · AnonReviewer1 · 2018-11-12
**Unclear details, unconvincing experiments**

**Rating:** 4
**Confidence:** 3

**Review:**

This paper focuses on fast adaptation to new behaviour of the other agents of the environment, be it opponents or allies. To achieve this, a method based on MAML is proposed, with two main components:
1) Learn a model of some characteristics of the opponent, such as "the final goal, next action, or any other character we wish to predict"
2) Learn a policy that takes as input the output of the model and the state, and that outputs the action of the agent.

The goal is that after a phase of meta learning, where the agents learns how to play against some new agents sampled from the distribution of opponents, it can quickly adapt to a new unseen agent. ("Experimental results show that the agent can adapt to the new opponent with a small number of interactions with the opponent")

While the motivation of this work is clear and the goal important for the RL community, the experiments fail to support the claim above.

The first task they demonstrate their approach on is a chasing game, where the opponent has a private goal cell it tries to reach, and the agent has to chase it. At the end of the game, it gets a reward of 10 if it is on the same cell, 5 if in an adjacent cell, and 0 otherwise. The exact details of the dynamic are not really clear, for example what happens in the event of a collision is not mentionned, and the termination condition is not mentionned either. (the text reads "One game lasts for at least 15 steps", maybe it was meant to be "at most 15 steps" ?).
The first incoherent aspect of this experiment is that they use 800 iterations of meta-learning, and then, when testing, they fine-tune their networks against each test opponent during 4000 games. That is, they use 5 times more game when fine-tuning as opposed to when pre-training, which contradicts the claim "the agent can adapt to the new opponent with a small number of interactions with the opponent" (this is not really few-shot learning anymore).
Further more, they compare their approach with various ablations of it: they either remove the meta-learning for the model (MA), for the policy (MO), or both (NM). The description of the NM baseline is not very precise, but it seems that it simply boils down to a regular (dueling) DQN: In this setting, since the opponent appears to have a fixed goal, finetuning against a single opponent simply boils down to learning a policy that reaches a specific cell of the grid, which we can expect DQN to solve perfectly on a 8x8 grid with 4000 training games. And yet, the curves for NM in graph 2c is not only really noisy, but also falls far from the optimum, which the authors don't discuss. There might be a problem with the hyperparameters used or the training loop.

The second task is a blocking game: the opponent has to choose amongst 5 paths to get to the top, and the agent has to choose the same path in order to block it. The action space should be precisely described, as it stands it is difficult to understand the dynamic. There are at least two possible ways to parametrize the actions:
1) Similarly to the blocking game, the agents could move in the 8 directions. In that case, based on the picture 3a, it seems that the agent can just mirrors the move of the opponent: since the moves are simultaneous, that would mean that the agent is always one step late, but each path is long enough for the agent to reach the exit before its opponent (it explicitly stated that the agent needs to block the exit, and that the opponent will not change path during one game). That would imply that perfect play is possible without any meta-learning or oponent modeling, and once again the NM baseline (or any vanilly DQN/Policy gradient method) should perform much better.
2) One other alternative is to have an action space of 5 actions, which correspond to the 5 paths. In that case the game boils down to a bandit, since both agents only take one action. Note that under this assumption, the random policy would get the right path (and reward +10) with probability 1/5 and a wrong one (reward -10) with probability 4/5, which leads to an expectated reward of -10*4/5 + 10/5 = -6. This is not consistent with the graph 3c, since at the beginning of the training, the NM agent should have a random policy, and yet the graph reports an average reward of -10 (the -6 mark seems to be reached after ~1000 episodes)

The last task boils down to one opponent that reaches one cell on the right, and the agent must reach the matching cell on the left. In this setting, the same discussion on the action space as the second task can be made. We note that the episode for 16 steps, and the distance from the center to any cell is at most 4 steps: an optimal policy would be to wait for 4 steps in the middle, and as soon as the opponent has reached its goal, use the remaining 12 steps to get to the mirror one. Once again, this policy doesn't require any prediction on the opponent's goal, and it's hard to believe that DQN (possibly with an lstm) is not able to learn that near perfectly.


In a last test the authors compare the performance of their algorithms in a one shot transfer setting: they sample 100 opponents for each task and play only one game against it (no fine-tuning). It is not clear whether special care has been taken to ensure that none of the sampled opponents has already been seen during training.
We note that the rewards reported for MO and MA (resp 0.0 and -0.08) are not consistent with the description of the reward function: on the worst case, the opponent chooses a goal on one extreme (say y1 = 1) and the agent chooses an object on the other end (say y2 = 7). In that case, the reward obtained is sampled from a gaussian with mean \mu = 10 - 3/2 * |y1 - y2| (which in this case evalutes to 1), and variance 1. This is highly unlikely to give such a low average reward over 100 episodes (note that this is worst case, if the opponent's goal is not on the extreme, the expected reward is necessarily higher). One possibility is that the agent never reaches an object, but in that case it would imply the that the meta-learning phase was problematic.
We also note that it is explicited that the MOA, MO and MA methods are tested after meta-training, but nothing is precised for NM. Has it been trained at all? Against which opponents? Is it just a random policy? There are too many missing details for the results to be interpretable.


Apart from that, the paper contains a significant amount of typos and gramatical mistakes please proof-read carefully. Some of them are:
"To demonstrate that meta-learning can do take"
"player 1 is the red grid and player 1 is the green one"
"we further assume that there exist a distribution"
" the goal’s location over the map is visualize in figure"
"Both players takes actions simultaneously"

---

### Meta-Review · Area_Chair1 · 2018-12-15
**Interesting approach to sample-efficient MARL, but the experiments do not support the claims**

**Confidence:** 5
**Recommendation:** Reject

**Metareview:**

The paper extends MAML so that a learned behavior can be quickly (sample-efficiently) adapted to a new agend (allied or opponent). The approach is tested on two simple tasks in 2D gridworld environments: chasing and path blocking.

The experiments are very limited, they do not suffice to support the claims about the method. The authors did not enter a rebuttal and all the reviewers agree that the paper is not good enough for ICLR.